# Tubular Dentin Regeneration Using a CPNE7-Derived Functional Peptide

**DOI:** 10.3390/ma13204618

**Published:** 2020-10-16

**Authors:** Yoon Seon Lee, Yeoung-Hyun Park, Dong-Seol Lee, You-Mi Seo, Ji-Hyun Lee, Joo-Hwang Park, Han-Wool Choung, So-Hyun Park, Won Jun Shon, Joo-Cheol Park

**Affiliations:** 1Laboratory for the Study of Regenerative Dental Medicine, Department of Oral Histology—Developmental Biology, School of Dentistry and Dental Research Institute, BK 21, Seoul National University, Seoul 08826, Korea; seon1663@snu.ac.kr (Y.S.L.); pyh5436@snu.ac.kr (Y.-H.P.); iceburge@snu.ac.kr (D.-S.L.); 2Regenerative Dental Medicine R and D Center, HysensBio Co., Ltd., Seoul 03080, Korea; seoym777@snu.ac.kr (Y.-M.S.); jhlee@hysensbio.com (J.-H.L.); orange@hysensbio.com (J.-H.P.); 3Department of Oral and Maxillofacial Surgery, School of Dentistry and Dental Research Institute, Seoul National University, Seoul 03080, Korea; woolmania@naver.com; 4Department of Conservative Dentistry, School of Dentistry and Dental Research Institute, Seoul National University, Seoul 03080, Korea; seon1663@naver.com (S.-H.P.); endoshon@gmail.com (W.J.S.)

**Keywords:** odontoblast, dentinogenesis, biomaterial, mineralized tissue development, dentin

## Abstract

We aim to examine the effects of a newly developed peptide derived from CPNE7 (Cpne7-DP) in tertiary dentin formation and peritubular space occlusion, and comprehensively evaluate its potential as a bioactive therapeutic agent. Human dental pulp cells (HDPCs) and a mouse pre-odontoblast cell line, MDPC-23, were chosen for in vitro studies to characterize lineage-specific cell responses after Cpne7-DP treatment. Whether Cpne7-DP reproduces the dentin regenerative potential of CPNE7 was tested using a beagle dog model by generating dentinal defects of various degrees in vivo. Peritubular space occlusion was further examined by scanning electron microscopy and microleakage test, while overall mineralization capacity of Cpne7-DP was tested ex vivo. CPNE7 promotes tubular dentin formation under both shallow and deep dentinal defects, and the functional peptide Cpne7-DP induces odontoblast-like differentiation in vitro, mineralization ex vivo, and tubular dentin formation in in vivo beagle dog dentin exposure and pulp exposure models. Moreover, Cpne7-DP leads to peritubular space occlusion and maintains stability under different conditions. We show that CPNE7 and its derivative functional peptide Cpne7-DP promotes dentin regeneration in dentinal defects of various degrees and that the regenerated hard tissue demonstrates the characteristics of true dentin. Limitations of the current dental materials including post-operative hypersensitivity make biological repair of dentin a field of growing interest. Here, we suggest that the dual functions of Cpne7-DP in tubular dentin formation and peritubular space occlusion are promising for the treatment of dentinal loss and sensitivity.

## 1. Introduction

Loss of tooth dentin not only generates unpleasant pain but ultimately leads to the weakening of whole tooth stability due to reduced dentin thickness. Among its varying causes, dental caries is the most prevalent chronic disease in both children and adults, affecting more than 5 billion people worldwide. Dental caries is caused by bacterial infection and resulting acidic byproducts that demineralize the tooth. Depending on the extent of the infection, the carious lesion either stops at the enamel or penetrates into the dentin [1]. Exposed dentinal tubules in dental caries serve as a route for bacterial invasion that may result in various pulpal responses [2]. In severe cases, tooth vitality is permanently lost [3]. A healthy tooth is composed of nearly 70% dentin enclosing the entire dental pulp, which is a pool of diverse stem cells [4]. Loss of tooth dentin not only generates unpleasant sensitivity but ultimately leads to the weakening of whole tooth stability due to reduced dentin thickness. Accordingly, incessant efforts have been made to regenerate biologic dentin. However, most of the previously reported reparative tertiary dentin shows features of osteodentin, which is composed of bone-like structures with osteocyte-resembling cells entrapped in lacunae-like spaces [5]. The formation of osteodentin remains valuable because the damaged tissue is repaired with a newly mineralized barrier. Nonetheless, many reports highlight the significance of dentinal tubules in terms of defense mechanisms [6,7,8]. Moreover, the mineral density of osteodentin is less than that of physiologic tubular dentin [9]. Therefore, the identification of biological materials capable of restoring dentin with tubular architecture is clinically important.

Based on the knowledge that epithelial–mesenchymal interaction is essential for tooth development, we previously discovered a dental epithelium-derived protein called Copine 7 (CPNE7). CPNE7 is an evolutionarily conserved, calcium-dependent phospholipid-binding protein which consists of two C2 domains in the N terminus and a von Willebrand factor A domain in the C terminus [10]. Secreted from pre-ameloblasts, CPNE7 induces odontoblast differentiation in vitro and promotes dentin formation ex vivo [11,12,13]. As a result, CPNE7 was suggested as a new molecule with the potential to diffuse across the dentin and induce tertiary dentinogenesis [14]. The fact that recombinant CPNE7 is a cell-derived soluble bioactive molecule makes it a promising candidate for use in regenerative dental medicine.

As no peptide has yet been reported to form physiologic tubular dentin by stimulating odontoblasts or odontoblastic differentiation, we sought to design and synthesize a stable, cost-efficient, and manipulative functional peptide that mimics the functions of the protein CPNE7. Here, we thoroughly evaluate the tubular dentin forming capacity of CPNE7-derived oligopeptide (Cpne7-DP) and suggest their use as future therapeutic agents for dentinal defects such as dental caries.

## 2. Materials and Methods

### 2.1. Tooth Defect Models with Canine Teeth

For in vivo canine studies, a total of 10 beagle dogs (2 y) were operated on, with all of them reaching end points. For each animal, 4 to 6 maxillary premolars and 6 mandibular premolars were used, depending on their periodontal conditions. Control and experimental groups were allocated to each and every animal so that possible biases resulting from individual nutritional, behavioral, and pulpal conditions could be eliminated. Compared to small animal studies, the sample size was necessarily limited for feasibility and ethical reasons. While the exact numbers for each experiment are included in the figure legends, at least 6 premolars and up to 12 premolars were assigned to every control and experimental group. Investigators performing beagle operation and sampling were not blinded, while investigators involved in histological analysis were blinded. All experiments using animals followed protocols approved by the Institutional Animal Care and Use Committee of Seoul National University (SNU-180416-2-1 and SNU-171020-5-2).

Beagle dogs of 1–2 years of age were used for three independent experiments. After disinfecting the cervical regions of the maxillary and mandibular premolars with 0.5% chlorhexidine, dish-shaped class V cavities were prepared using a #4 high-speed round bur (head diameter, 1.4 mm). For shallow cavities, the drilling stopped when half to two-thirds of the bur head penetrated the tooth structure, depending on the tooth size. For deep cavities, the drilling stopped when the color of the remaining dentin looked reddish gray. The smear layer was removed with a gentle application of 17% EDTA. After sufficient irrigation, surgical sites were dried with a cotton pellet. For CPNE7 function analysis, the cavities were either untreated or treated with a topical application of rCPNE7 (recombinant CPNE7) protein (1 µg total rCPNE7 per tooth in a buffer containing 25 mM Tris-HCl, 100 mM glycine, and 10% glycerol). After a brief incubation for diffusion, the cavities were then filled with either glass ionomer (GI) cement (GC; Fuji II LC; GC America Inc., Alsip, IL, USA), composite resin (Filtek Supreme Ultra Flowable Restorative; 3 M, MN, USA) or intermediate restorative material (IRM; Dentsply Sirona, New York, PA, USA). For peptide function analysis, an HIV-1 Tat-derived cell-penetrating peptide was used as control. In addition to the abovementioned shallow and deep cavity models, complete pulp exposure was performed for peptide analysis. For complete pulp exposure, the drilling stopped when a pin-point exposure of the pulp was generated and bleeding was evident. Cotton pellets soaked in saline were used for bleeding control. Cavities with exposed pulps were divided into two groups for the experiment. Group 1 received only GI cement filling, group 2 received GI cement filling after MTA (ProRoot MTA; Dentsply Sirona, New York, PA, USA) sealing, and group 3 received GI cement filling after MTA mixed with CPNE7-derived peptide (Cpne7-DP) sealing.

The premolar areas were dissected 3, 6, or 12 weeks following surgery, and samples were immersed in 4% paraformaldehyde and kept for an additional 24 h at 4 °C. Decalcification was performed in 10% formic acid, and the specimens were then embedded in paraffin. Serial 5-µm-thick sections were H&E stained.

### 2.2. TUNEL Assay

Apoptotic cells were detected using the TUNEL kit (Roche Biochemicals, Basel, Switzerland), according to the manufacturer’s instructions. Endogenous peroxidase activity within the tissue sections were blocked in 3% H_2_O_2_ prior to enzymatic labeling. To yield a colored reaction product, enzymatically labeled cells were then incubated with 3,3-diaminobenzidine tetrahydrochloride. Visualization was achieved by incubating the sections with diaminobenzidine tetrahydrochloride (DAB), and subsequent counter-staining with hematoxylin. Positive TUNEL signals were converted to red color using IHC Profiler of Image J software (National Institute of Health, Gaithersburg, MD, USA).

### 2.3. Cell Culture

MDPC-23 cells were provided by Dr. J.E. Nor (University of Michigan, Ann Arbor, MI, USA) and cultured in Dulbecco’s modified Eagle medium (DMEM; Gibco BRL., Carlsbad, CA, USA). C3H10T1/2 cells were obtained from the American Type Culture Collection (ATCC, Manassas, VA, USA) and cultured in RPMI 1640 medium (Gibco BRL). Both cell lines were supplemented with 10% heat-inactivated fetal bovine serum (FBS; Gibco BRL) and antibiotic-antimycotic reagents (Gibco BRL) at 37 °C in a 5% CO_2_ atmosphere. Impacted human third molars from patients between the ages of 18 and 22 were provided by the Seoul National University Dental Hospital. The experimental protocol was approved by the Institutional Review Board (IRB No: S-D20140007). Informed consent was obtained from every patient. Isolation of whole pulp cells was performed as previously described [11], and cells were cultured in minimum essential media α (MEM-α; Gibco BRL) for use in in vitro and ex vivo experiments. For human dental pulp cells (hDPCs) and MDPC-23 cell differentiation, 80–90% confluent cells were cultured in corresponding media supplemented with 5% FBS, ascorbic acid (50 µg/mL), and β-glycerophosphate (10 mM) for up to 3 weeks. Passages from 2 to 4 were used for hDPSCs, and 23 to 25 were used for MDPC-23 cells.

### 2.4. Peptide Synthesis

Cpne7-DP consists of a synthetic peptide corresponding to the 10 amino acid residue 344–353 fragment (KYKQKRRSYK) of the hCPNE7 protein. The peptides were synthesized using the Fmoc (9-fluorenylmethoxycarbonyl)-based solid-phase method and characterized by Lugen Sci. Co., Ltd. (Bucheon, Korea). The purity of the peptides used in this study was greater than 97%, as determined by high-performance liquid chromatography.

### 2.5. Luciferase Assay

MDPC-23 cells were seeded on a 24-well plate at a density of 5 × 10^4^ cells/wells and transfected with Lipofectamine Plus™ reagent (Invitrogen, Carlsbad, CA, USA) after 24 h. For each transfection, 0.4 µg luciferase reporter plasmid pGL3basic (Control) and pGL3LUC *dspp* (−750~61) were used. Transfected cells were treated with Cpne7-DP or recombinant CPNE7 protein for 48 h and were then lysed for luciferase activity assessment using the luciferase reporter gene assay system (Roche Applied Science, Indianapolis, IN, USA) according to the manufacturer’s instructions. The measurements were performed with a luminometer (FLUOStar OPTIMA, BMC Laboratory, Offenburg, Germany). Three independent experiments with triplicate samples were analyzed.

### 2.6. Cytotoxicity Assessment

To evaluate the effects of Cpne7-DP on hDPCs proliferation, an MTT [3-(4,5-Dimethylthiazol-2-yl)-2,5-diphenyltetrazolium bromide, a tetrazole] assay (Sigma-Aldrich, Saint Louis, MO, USA) was performed. Cells were seeded on a 96-well plate at a density of 3 × 10^3^ cells/well and treated with Cpne7-DP after 24 h. Cells were cultured for up to 5 days, and MTT analyses were performed on days 0, 1, 3, and 5. After washing with phosphate-buffered saline (PBS), 20 µL of MTT was added to each well and incubated for 4 h at 37 °C. After removing the MTT solution, the converted dye was dissolved in Me_2_SO and measured by reading the absorbance at a wavelength of 540 nm with a microplate reader (Multiskan EX; Thermo Electron Corp., Waltham, MA, USA). Three independent experiments with triplicate samples were analyzed.

### 2.7. Real-Time Polymerase Chain Reaction Analysis

Total RNA was extracted from cells with TRIzol reagent according to the manufacturer’s instructions (Invitrogen, Carlsbad, CA, USA). Three µg of RNA was reverse transcribed using Superscript III reverse transcriptase (Invitrogen) and oligo (dT) primers (Invtrogen). One µL of cDNA was subjected to PCR amplification using the ABI PRISM 7500 sequence detection system (Applied Biosystems, Carlsbad, CA, USA) using SYBR Green PCR Master Mix (Applied Biosystems) according to the manufacturer’s instructions. PCR was performed using the following conditions: 94 °C for 1 min, 95 °C for 15 s, and 60 °C for 34 s, for 40 cycles. All reactions were performed in triplicate and normalized to reactions using the housekeeping gene glyceraldehyde 3-phosphate dehydrogenase (GAPDH). Relative changes in gene expression were calculated using the comparative threshold cycle (CT) method. The sequences of the real-time PCR primers used in the study are listed in Table 1.

### 2.8. Western Blot Analysis

Whole cell lysates were harvested in a lysis buffer consisting of 50 mM Tris-HCl, pH 7.4, 150 mM NaCl, 1% Nonidet P-40, 1 mM EDTA, and 1 mM PMSF supplemented with protease inhibitors (Roche Molecular Biochemicals, Mannheim, Germany). The supernatant after centrifugation at 13,000× *g* for 30 min was collected for analysis. DCTM protein assay system (Bio-Rad Laboratories, Hercules, CA, USA) was used to measure protein concentrations. Twenty-five μg of proteins were resolved using 10% polyacrylamide gel electrophoresis and transferred to a PVDF membrane, which was then blocked with PBST (10 mM phosphate-buffered saline, pH 7.0, and 0.1% Tween-20) buffer containing 5% non-fat dry milk for 1 h at room temperature. After washing, the blots were incubated with primary antibodies indicated in Table 2, overnight at 4 °C with gentle shaking. Before incubation with anti-rabbit or anti-mouse immunoglobulin G conjugated to horseradish peroxidase in PBST for 1 h at room temperature, blots were washed 3 times for 10 min in PBST. Labeled protein bands were detected under an enhanced chemi-luminescence reagent (ECL; Santa Cruz Biotechnology, Dallas, TX, USA) according to the manufacturer’s guidelines. Semi-quantitative analyses were performed using Image J software (National Institute of Health).

### 2.9. Ex Vivo Transplantation and Histological Analysis

Human root segments (<5 mm in thickness) were prepared from extracted human teeth after pulp tissues were removed according to our previously reported protocol [15]. Human DPCs (2 × 10^6^) were mixed with hydroxy apatite/tricalcium phosphate (HA/TCP) ceramic powder (Zimmer, Warsaw, IN, USA) alone or with Cpne7-DP (10 µg) in an 0.5% fibrin gel and then transplanted subcutaneously into immunocompromised mice (NIH-bg-nu-xid; Harlan Laboratories, Indianapolis, IN, USA) for 6 or 12 weeks. Dentin/pulp-like tissue formation in the emptied root canal space was evaluated after mixing hDPCs (2 × 10^6^) with Cpne7-DP (10 µg) in a 0.5% fibrin gel and inserting the mixture into emptied root canal space of the human root segments for 6 weeks. Harvested samples were fixed in 4% in paraformaldehyde, decalcified in 10% EDTA (pH 7.4), embedded in paraffin, and stained with hematoxylin and eosin (H&E) (Vector Laboratory, Burlingame, CA, USA), Masson’s Trichrome Stain (Polysciences, Warrington, PA, USA), or processed for immunohistochemical analysis. For immunohistochemistry, the sections were incubated overnight at 4 °C with rabbit polyclonal DSP and BSP produced, as described previously [16] at a dilution of 1:150. Biotin-labeled goat anti-rabbit IgG (Vector Laboratory) was incubated with the sections at room temperature for 30 min, which were then reacted with the avidin-biotin-peroxidase complex (Vector Laboratory). Signals were converted using a diaminobenzidine kit (Vector Laboratory). Nuclei were stained with hematoxylin.

### 2.10. Immunofluorescence Staining

For peptide translocation and localization analysis, a Cy5 Fast Conjugation Kit (ab188288, Abcam, Cambridge, MA, USA) was used to tag Cpne7-DP according to the manufacturer’s instructions. MDPC-23 cells were treated with Cy5-labeled Cpne7-DP (10 µg/mL) and fixed 3 h later. Before fixing with 4% paraformaldehyde in PBS, cells in Laboratory-Tek chamber slides (Nunc, Rochester, NY, USA) were washed with PBS. Cells were then visualized using fluorescence microscopy (AX70, Olympus, Tokyo, Japan). DAPI (Sigma-Aldrich) was used to identify the chromosomal DNA in the nucleus (1:1000 dilution). Reagents used are listed in Table 3.

### 2.11. Transient Transfection

C3H10T1/2 cells or hDPCs were seeded on 60 mm culture plates at a density of 1.0 × 10^6^ cells. The cells were transiently transfected with DDK (Flag)-tagged CPNE7 using the Metafectene Pro reagent (Biontex Laboratories GmbH, Munich, Germany) according to the manufacturer’s instructions.

### 2.12. Scanning Electron Microscopic Analysis

Samples were fixed in 0.1 M cacodylate buffer (pH 7.3) containing 2.5% glutaraldehyde for 30 min and in 0.1 M cacodylate buffer (pH 7.4) containing 1% osmium tetroxide for 1 h. Samples were then rapidly dehydrated through an ethanol gradient. After sputter coating with gold, samples were observed under a scanning electron microscope (S-4700, HITACHI, Tokyo, Japan).

### 2.13. Specimen Preparation

Human third molars were collected at the Seoul National University Dental Hospital (Seoul, Korea), and the experimental protocol was approved by the Institutional Review Board (IRB No: S-D20140007). Teeth were decoronated to 2–3 mm using a safe diamond disc (Isomet, Buehler Ltd., Düsseldorf, Germany) to eliminate the coronal enamel layer and expose the dentin surface. Teeth were then treated with 5 mL of 32% phosphoric acid solution for 5 min to completely open the dentinal tubules. To remove the residual smear layer, specimens were ultrasonicated in deionized water twice for 5 min each and then rinsed with PBS three times.

### 2.14. CLSM Specimen Preparation and Analyses

Cpne7-DP was tagged with Rhodamin B dye using the Rhodamine Fast Conjugation kit (ab188286, Abcam, Cambridge, MA, USA) according to the manufacturer’s instructions. Rhodamin B-labeled Cpne7-DP (20 µg) was applied to the dentin surfaces of specimens with the help of disposable microbrushes for 1 min, and then specimens were washed with PBS three times. Subsequently, the specimens were longitudinally sectioned at a thickness of 0.5 mm using a safe diamond disc. The sections were mounted onto glass slides and scanned under a confocal laser scanning microscope (LSM 700; Carl Zeiss, Jena, Germany).

### 2.15. Alizarin Red S Staining

MDPC-23 cells were seeded on 60 mm culture plates at a density of 1.0 × 10^6^ cells and cultured in differentiation medium for 14 days with or without Cpne7-DP. The formation of mineralized nodules was evaluated by staining with alizarin red S (Sigma-Aldrich) solution in 0.1% NH_4_OH at pH 4.2 for 20 min at room temperature.

### 2.16. Microleakage Test

The extent of tubule occlusion was measured with the previously reported nano-fluid movement measuring device [17]. Apical 3 mm of extracted beagle incisors was cut with a high-speed diamond bur (TF-13, MANI, Tokyo, Japan) to expose the root canals. A 0.9 mm metal tube was inserted, while 32% phosphoric acid, adhesive agent (Singlebond Universal, 3M ESPE, St Paul, MN, USA) and flowable composite resin (Filtek Supreme Ultra Flowable Restorative; 3M, Alexandria, MN, USA) were applied to bond with the metal tube. All areas except for the defect region were covered with nail varnish several times. The prepared specimens were kept in distilled water. A nano-fluid movement measuring device (NanoFlow, IB Systems, Seoul, Korea) recognizes the movement of bubbles due to leakage when distilled water is left to flow from the tooth apex to the exposed dentin at 70 cm H_2_O. All measurements were taken 40 min after the specimens were connected, while the first 20 min of outflow was excluded.

### 2.17. Statistical Analysis

All data were expressed as the mean ± standard deviation of triplicate experiments. Statistical significance was analyzed using a non-parametric Mann–Whitney U test for comparisons between two groups, and one-way analysis of variance (ANOVA) with Bonferroni correction for comparisons between more than two groups by the SPSS software version 25. *p* values less than 0.05 were considered statistically significant.

## 3. Results

### 3.1. Treatment of Recombinant CPNE7 Promotes the Regeneration of Tubular Dentin in Shallow and Deep Cavity Models

To assess the effects of recombinant CPNE7 (rCPNE7) in varying extents of dentinal defects, both shallow and deep cavities were artificially generated in beagle dog premolars (Figure 1a) and divided into two groups each (*n* = 6 for all groups): glass ionomer (GI), cement filling only (Control), and GI cement filling after topical application of rCPNE7. After three weeks, in the shallow cavity models where the underlying odontoblasts were expected to be mildly deterred, no new hard tissue formation was detected in the pulp cavity of the control group. In contrast, newly generated tertiary dentin retaining the physiologic tubule structure was observed beneath the defect in the rCPNE7-treated group (Figure 1b). In the deep cavity models where the underlying odontoblasts were expected to be severely damaged, no evidence of dentin repair was found in the control group. Moreover, the odontoblasts beneath the cavity became more rounded than the adjacent odontoblasts underlying the non-cavity areas, indicating morphological distortion. In the rCPNE7-treated deep cavity, however, newly generated tertiary dentin was present beneath the remaining dentin at the cavity-preparation sites with regular cell alignment (Figure 1c). Terminal deoxynucleotidyl transferase dUTP nick end labeling (TUNEL) demonstrated significantly reduced cell death under the newly generated dentin in the rCPNE7-treated group (Figure 1d). When the samples were collected six weeks after the experiment instead of three, the thickness of the newly formed tubular dentin increased significantly in both shallow and deep cavity models (Appendix A
Figure A1a,b).

We next investigated whether the CPNE7-induced hard tissue retains the characteristics of physiologic dentin. The presence of dentinal tubules in the newly mineralized area was examined by scanning electron microscopic analysis, and a clear tubule structure was identified along the entire length of predentin (Figure 1e). Masson’s trichrome staining revealed that the newly generated dentin area contained a higher collagen content, indicative of recent mineralization activity (Figure 1f). The cytoplasmic regions of odontoblast processes were also distinctly stained within the newly formed dentin. To eliminate the possibility of restorative materials affecting the action of CPNE7, we used composite resin or zinc oxide-eugenol (ZOE) as an alternative cavity filling material. Similarly, CPNE7 treatment led to the formation of tubular dentin, demonstrating its regenerative potential regardless of the restorative material used (Figure A2).

### 3.2. Functional Peptide, Cpne7-DP Directly Penetrates Odontoblastic Cells and Upregulates Odontoblast Differentiation Markers In Vitro

We designed and synthesized an oligopeptide covering the linker region of CPNE7. The amino acid sequence of the oligopeptide was “Lys-Tyr-Lys-Gln-Lys-Arg-Arg-Ser-Tyr-Lys,” and was named as Cpne7-DP (Figure 2a). Cpne7-DP was localized to the cytoplasm, and there was no evidence of nuclear labeling (Figure 2b). MTT assay results confirmed that there was no significant difference in cytotoxicity from 0 to 5 days between control and Cpne7-DP -treated groups (Figure 2c). The peptides were diffusely located in the cytoplasm of *nucleolin* siRNA-transfected cells similar to control, suggesting that Cpne7-DP does not require nucleolin as its receptor (Figure 2d). Next, we treated MDPC-23 cells with chlorpromazine (CPZ), an inhibitor of clathrin-mediated endocytosis, or methyl-β-cyclodextrin (MβCD), an inhibitor of caveolin-mediated endocytosis, and examined the localization of Cpne7-DP by immunofluorescence. Cpne7-DP diffusely localized to the cytoplasm in both groups (Figure 2e,f). Thus, we concluded that Cpne7-DP does not act through receptor-mediated endocytosis. Treatment with the macropinocytosis inhibitor ethylisopropyl amiloride (EIPA) also showed no effect on the cytoplasmic localization of Cpne7-DP, suggesting that Cpne7-DP is able to directly penetrate odontoblastic cells (Figure 2g).

The expression levels of odontoblast marker genes, dentin sialophosphoprotein (dspp), dentin matrix protein 1 (DMP1), and Nestin, increased in hDPCs treated with Cpne7-DP (Figure 3a). Cpne7-DP also elevated the DSP expression level compared to control (Figure 3b). Next, we conducted luciferase reporter assays using a dspp-responsive reporter in mouse dental papilla-derived MDPC-23 cells treated with either rCPNE7 or Cpne7-DP. Treatment of Cpne7-DP increased *dspp* promoter activity as much as rCPNE7 in a dose-dependent manner, compared to control (Figure 3c). To determine the extent to which Cpne7-DP stimulates mineralization, we examined the mineralization capacity of Cpne7-DP on odontoblast-like cells in vitro. Alizarin red S staining results indicated that Cpne7-DP -treated MDPC-23 cells began to demonstrate mineralized nodule formation on day 7, which significantly increased until day 14. The control group showed fewer mineralized nodules than the Cpne7-DP-treated group microscopically (Figure 3d).

### 3.3. Functional Peptide, Cpne7-DP, Promotes Dentin-Like Tissue Formation Ex Vivo

We further evaluated the role of Cpne7-DP by transplanting hDPCs into the subcutaneous tissues of immunocompromised mice using hydroxyapatite/tricalcium phosphate (HA/TCP) under three different conditions: hDPCs only (Control), hDPCs treated with Cpne7-DP, and hDPCs treated with recombinant bone morphogenetic protein 2 (rBMP2). Twelve weeks after the transplantation, the Cpne7-DP treatment group manifested the formation of dentin-pulp-like tissue with cells inserting long cellular processes into the tubule-like structure formed within the newly mineralized tissue. In comparison, the rBMP2 treatment group formed the typical bone-like structure with round-shaped cells entrapped within. Moreover, the formation of bone-marrow-like structures was evident inside the mineralized tissue of the rBMP2 treatment group (Figure 3e and Figure A3a). We next checked the presence of dentin sialoprotein (DSP) and bone sialoprotein (BSP), the respective dentin and bone marker proteins by immunohistochemical staining. Whereas mineralized tissues in the rBMP2 treatment group showed strong BSP and much weaker DSP expression, those in the Cpne7-DP treatment group showed strong DSP staining (Figure A3b). The odontoblast-like cells lining the newly formed tissue in the Cpne7-DP treatment group were also highly positive for DSP. Cell morphologies observed with scanning electron microscopy demonstrated the development of long process-like structures in the Cpne7-DP treatment group (Figure A3c). We additionally tested the function of Cpne7-DP by subcutaneous transplantation of hDPCs with or without Cpne7-DP treatment into the emptied human root canal space. Twelve weeks after the transplantation, newly formed dentin-like structures with occasional cell entrapments were observed in the Cpne7-DP treatment group. Moreover, a considerable number of dentinal tubules were occupied by transplanted cells exhibiting elongated phenotypes (Figure A3d). The scanning electron microscope images of the pulp–dentin interface in the Cpne7-DP treatment group showed odontoblast-like cells embedding their cellular processes into the dentinal tubules (Figure A3e).

### 3.4. Cpne7-DP Promotes the Regeneration of Tubular Dentin and Dentinal Tubule Occlusion in Shallow and Deep Cavity Models

There has been no peptide reported, which can stimulate physiologic dentin regeneration in vivo. A dentin matrix protein-derived peptide, Cpne7-DP, was first subjected to the assessment of its ability to permeate dentinal tubules. Cpne7-DP labeled with a fluorescent dye was applied by microbrushing onto the pre-exposed dentinal tubules of an extracted tooth for 1 min. After moderate washing, the peptide was observed to flow through the entire length of the dentinal tubules into the pulp by confocal microscopy (Figure 4a). We further investigated its function in dentin regeneration in beagle dog tooth defect models. An HIV-1 Tat-derived cell-penetrating peptide was used as control, where its treatment led to no new hard tissue formation. In both shallow and deep cavity models, the treatment of Cpne7-DP resulted in the formation of tertiary dentin underneath the defect after 3 weeks, which retained the tubule structure continuous with that of the remaining dentin (Figure 4b). Furthermore, the thickness of the newly generated dentin was greater when samples were analyzed after 6 weeks (Figure A4).

Next, we analyzed peritubular dentin formation in beagle tooth defect models by scanning electron microscopy. After removing the filling material, the defect surface where dentinal tubules were exposed by tooth preparation was observed (Figure 4c). In the control group where defects received no treatment, a number of exposed tubule openings were seen. The Cpne7-DP treatment group, however, showed much smaller tubule openings, suggesting the possibility that underlying odontoblasts had formed dentin in peritubular areas as well as in the tubule spaces above their cellular processes. According to the nano-fluid movement test used in our previous study [18], the Cpne7-DP-treated group demonstrated a significantly smaller volume of liquid flow than the control group (Figure 4d). Finally, we cross-sectioned the tooth between the defect and the pulp space, and compared the sizes of the exposed tubule openings. As in the defect surface, dentinal tubules in the Cpne7-DP treatment group became generally narrower than those in the control group (Figure 4e).

### 3.5. Cpne7-DP Treatment Promotes Tubular Dentin Regeneration in Pulp Exposure Models

Finally, we generated a pulp exposure model to examine the effect of Cpne7-DP, where odontoblasts undergo consequential cell death. For more detailed histological analysis, it was further divided into three groups: GI cement filling only (Control), GI cement filling after mineral trioxide aggregate (MTA) sealing (MTA group), and GI cement filling after sealing with Cpne7-DP-mixed MTA (Cpne7-DP + MTA group). In the control group, no new hard tissue formation was found at the pulp exposure site, whereas, in the MTA group, a typical bone-like dentin bridge (osteodentin) was newly formed around the exposure site (Figure 5a). As in conventional direct pulp capping using MTA, the osteodentin bridge showed round-shaped cell entrapments within the mineralized tissue. Moreover, hardly any cell alignment was observed under the osteodentin bridge. In the Cpne7-DP + MTA group, on the other hand, dentin bridge with tubular architecture was generated, which had started to close the exposure site (Figure 5a). Furthermore, the palisade layer of odontoblasts expressing DSP and Nestin was well preserved and lined the dentin bridge (Figure 5b).

## 4. Discussion

Dentin has a number of similarities with bone both in its chemical composition and its mode of formation; however, it does not undergo lifelong remodeling and cannot be replaced after loss [19,20]. Odontoblasts are ectomesenchyme-derived post-mitotic cells that are responsible for dentin development and eventually line dentin [21]. Although odontoblasts have recently been reported to have sensory and immune cell capacities [22,23], their primary role is the secretion of the organic matrix that will be progressively mineralized [24]. Noticeably, odontoblasts elongate their cellular processes as they secrete and form predentin. These odontoblastic processes ultimately get embedded in the mineralized dentin matrix, establishing the dentinal tubule structure [25]. The secretory activity of a mature odontoblast that has completed primary and secondary dentin formation diminishes under healthy conditions. These resting state odontoblasts typically shrink in size and accumulate lipofuscin due to decreased autophagic activity [26]. In response to dentinal defects, either underlying odontoblasts are reactivated to generate physiologic reactionary dentin or other pulp cell sources are recruited and differentiate into odontoblast-like cells to produce pathologic reparative dentin [27]. These endogenous processes may be a key to regenerative medicine for tooth dentin.

Shallow and mild injuries to dentin stimulate the matrix-secreting activity of underlying resting state odontoblasts. The newly formed tertiary dentin is continuous with remaining dentin through the dentinal tubule structure and is called “reactionary” dentin [27]. In the case of deep and severe dentin injuries, underlying odontoblasts partially or massively undergo cell apoptosis, and a sub-odontoblastic population is recruited to differentiate into odontoblast-like cells [28,29,30]. Whether these pulp cells have really differentiated into odontoblasts can be examined by the morphology of the newly generated mineralized tissue called “reparative dentin.” In the present study, treatment of CPNE7 or its derivative peptide Cpne7-DP resulted in the regeneration of tubular dentin in both the shallow and deep cavity models, indicating their abilities to not only induce new odontoblast differentiation, but to reactivate the matrix-secretion of underlying odontoblasts.

In addition to causing hypersensitivity, exposed dentinal tubules can be a route for the invasion of foreign substances [31]. The current treatment for dentin hypersensitivity patients incorporates insoluble precipitate formation in the open tubules [32,33]. The ability of Cpne7-DP to stimulate underlying odontoblasts to resume matrix-secreting activity results in peritubular dentin formation. In addition to the newly generated tubular dentin along the pulp side, our SEM analysis revealed that peritubular dentin is deposited above the odontoblast process near the dentin-restoration interface. Consequently, it resulted in the physiologic occlusion of dentinal tubules. Integrity of the tubule occlusion was further confirmed by a nano-fluid movement test.

A number of synthetic peptides, especially those derived from proteins in an enamel or dentin matrix, have been developed to promote remineralization of enamel and dentin [34,35,36,37]. Most were capable of inducing hydroxyapatite deposition, and a DSP-derived peptide was reported to act on pulp cells and induce osteodentin formation [38]. In the present study, Cpne7-DP highly reproduced the in vitro effects of CPNE7 by upregulating odontoblast marker genes, *DSPP*, and *Nestin*. Moreover, subcutaneous transplantation of Cpne7-DP-treated hDPCs resulted in the formation of a dentin-pulp-like complex ex vivo. Unlike CPNE7, which is internalized via nucleolin-mediated endocytosis, Cpne7-DP seems to act on odontoblasts by directly penetrating the cell membrane. Exactly how odontoblast marker gene transcription is regulated by Cpne7-DP is an area of future exploration.

Although the efficiency of hard tissue regeneration differs in model animals, regeneration studies in large mammals (e.g., dogs) have been reported to be more challenging than in small animals (e.g., mice) [39,40]. Nevertheless, large mammals provide a more appropriate model for mimicking human disease and are thus more transferable to a human model. In our canine model, both CPNE7 and Cpne7-DP showed successful regeneration of tubular dentin. As the results were based on a single topical application, the effects of multiple repetitive applications should also be analyzed to determine their additive effects. The thickness of the newly formed tubular dentin increased with time, and 6-week samples demonstrated greater new dentin volume than 3-week samples. Such correlation implies that the secretory activity of either newly differentiated odontoblast-like cells or existing odontoblasts is prolonged once triggered. Nonetheless, one of the limitations of this study includes the lack of an exact quantification of newly formed tertiary dentin.

The findings of the current study suggest that Cpne7-DP promotes the formation of new tubular dentin by inducing odontoblast differentiation of dental pulp stem cells in both dentin and pulp exposure models (Figure 6, Table 4). The development of Cpne7-DP, a synthetic oligopeptide derived from CPNE7, proved to be advantageous not only in that it perfectly reproduces the functions of CPNE7, but it is a more stable and potent cell-penetrating peptide. Coherent observations in in vitro, ex vivo, and in vivo models establish the groundwork for the clinical translation of Cpne7-DP, which shows a promising ability to arrest the demineralization process, compensate for lost dentin in dentin defects, and occlude exposed dentinal tubules to reduce dentin hypersensitivity.

## 5. Conclusions

Regenerative dental medicine has rapidly progressed since the advancement of stem cell biology and material science. However, more emphasis has been placed on the success of regeneration than on how well the newly generated tissue retains the original organ structure and function. Especially in tooth regeneration research, the compositional similarity of bone and dentin has undermined their distinction. We show that CPNE7, a protein crucial for epithelial-mesenchymal interaction during tooth development, promotes dentin regeneration in dentinal defects of various degrees and that the regenerated hard tissue demonstrates the characteristics of true dentin. Moreover, a CPNE7 derivative synthetic oligopeptide was developed, which induces odontoblast differentiation in vitro and physiologic dentin formation in vivo. Comprehensive evaluation further validated its potential as a bioactive therapeutic agent. Our results suggest that the dual functions of CPNE7 and its peptide in tubular dentin formation and peritubular space occlusion are promising for oral disease-targeted application, especially those involving dentinal loss and pain.

## Figures and Tables

**Figure 1 materials-13-04618-f001:**
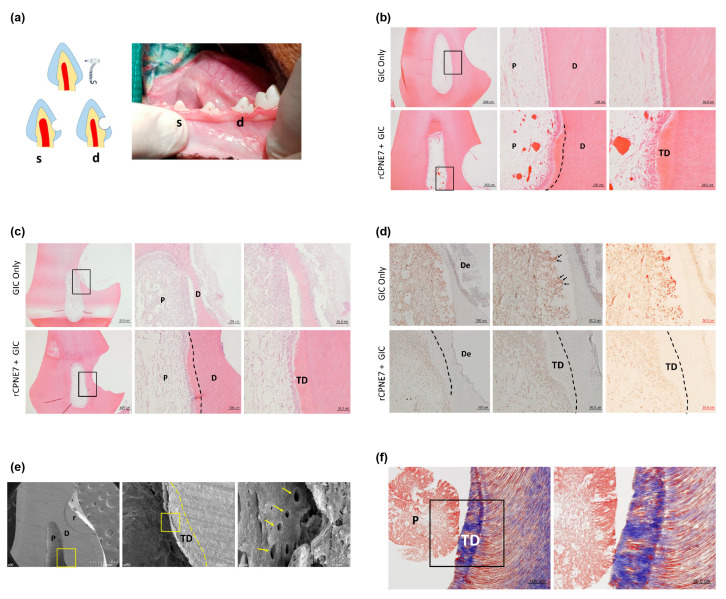
Treatment of rCPNE7 promotes the regeneration of tubular orthodentin in shallow and deep cavity models after three weeks. (**a**) Schematic illustration and representative macroscopic image of s) shallow and d) deep cavity prepared on the cervical areas of beagle dog premolars. (**b**) Histological analysis of dental pulp responses at the shallow cavity preparation areas treated with or without rCPNE7 by hematoxylin and eosin staining (*n* = 6). Dashed line indicates the remaining original dentin/newly formed dentin interfaces. (**c**) Histological analysis of dental pulp responses at the deep cavity preparation areas with or without rCPNE7 treatment by hematoxylin and eosin staining (*n* = 6). Dashed line indicates the remaining original dentin/newly formed dentin interfaces. (**d**) TUNEL staining of dental pulp at the deep cavity preparation areas with or without rCPNE7 treatment. (**e**) Scanning electron microscope images of the rCPNE7-treated group. Boxed areas indicate the dentin–pulp interface in the newly formed dentin. Arrows indicate the dentinal tubules; dashed line indicates the remaining original dentin/newly formed dentin interfaces. (**f**) Masson’s trichrome staining of newly formed dentin in rCPNE7-treated group. P, pulp; D, dentin; TD, newly formed tertiary dentin; De, defect; r, resin composite.

**Figure 2 materials-13-04618-f002:**
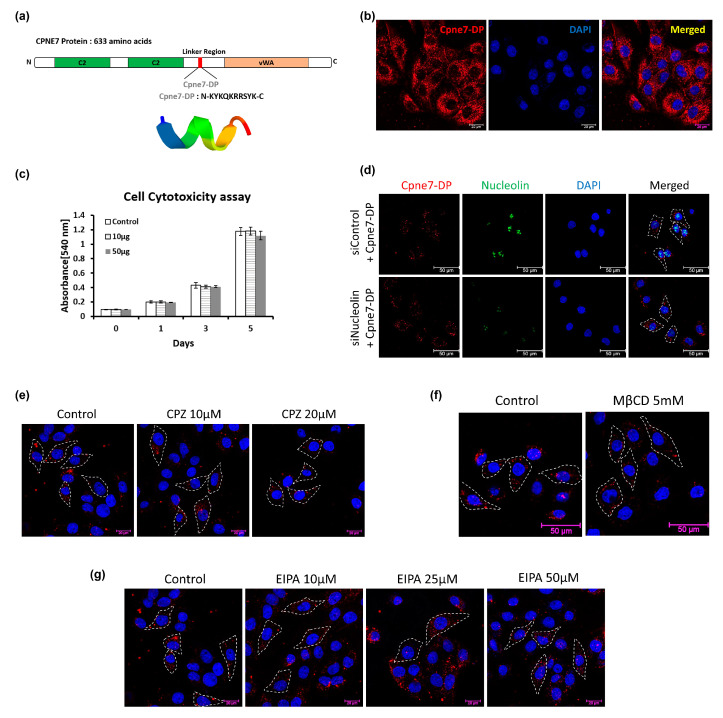
Functional peptide, Cpne7-DP directly penetrates odontoblastic cells. (**a**) Amino acid sequence and ribbon structure of Cpne7-DP derived from the linker region of CPNE7 protein. (**b**) Intracellular distribution of Cpne7-DP in odontoblastic MDPC-23 cells detected by immunofluorescence. (**c**) The effect of Cpne7-DP on the proliferation of hDPCs was evaluated using an MTT (3-(4,5-dimethylthiazol-2-yl)-2,5-diphenyl tetrazolium bromide) assay. (**d**) Internalization of Cpne7-DP detected by immunofluorescence in MDPC-23 cells after treatment with nucleolin siRNA. MDPC-23 cells were pre-treated with nucleolin siRNA for 1 h before Cpne7-DP treatment. (**e**) Internalization of Cpne7-DP detected by immunofluorescence in MDPC-23 cells pre-treated with varying concentration of chlorpromazine (CPZ; 10 and 20 μM) for 1 h before Cpne7-DP treatment. Dashed line indicates the cell edge. (**f**) Internalization of Cpne7-DP detected by immunofluorescence in MDPC-23 cells pre-treated with methyl-beta-cyclodextrin (MβCD) for 1 h before Cpne7-DP treatment. Dashed line indicates the cell edge. (**g**) Internalization of Cpne7-DP detected by immunofluorescence in MDPC-23 cells pre-treated with ethylisopropyl amiloride (EIPA) for 1 h before Cpne7-DP treatment. Dashed line indicates the cell edge. All values represent the mean ± standard deviation of triplicate experiments. DAPI, 4′, 6-diamidino-2-phenylindole; GAPDH, glyceraldehyde 3-phosphate dehydrogenase.

**Figure 3 materials-13-04618-f003:**
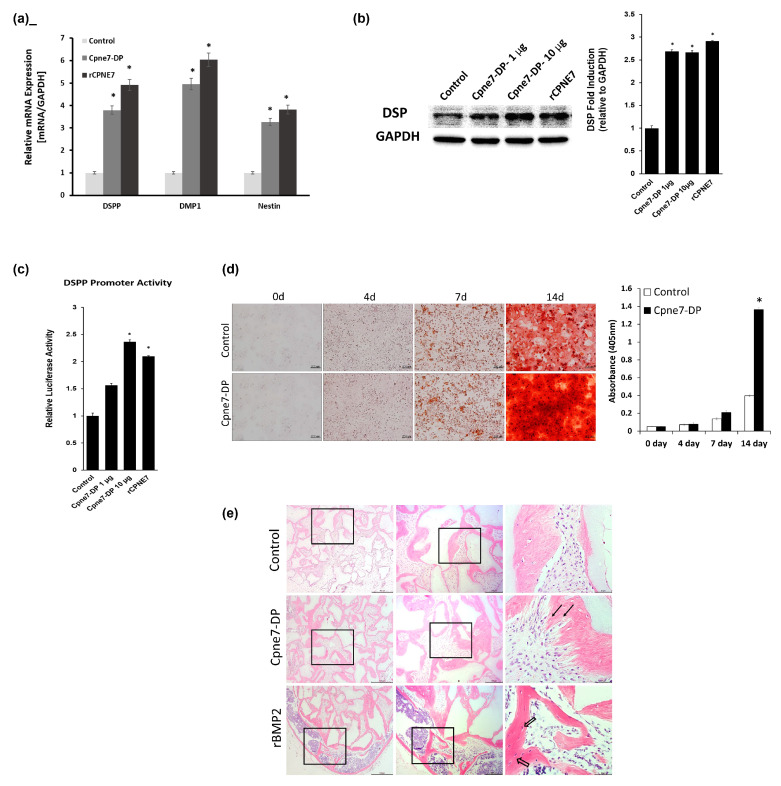
Functional peptide, Cpne7-DP upregulates odontoblast differentiation markers in vitro and induces dentin-like tissue formation ex vivo. (**a**) Real-time PCR analysis of DSPP, DMP1, and Nestin mRNA in hDPCs treated with Cpne7-DP or rCPNE7. (**b**) Western blotting and semi-quantification analysis of DSP expression in hDPCs treated with Cpne7-DP 1 µg, 10 µg or rCPNE7. (**c**) Transcriptional activity of dentin sialophosphoprotein (dspp) promoter was evaluated by luciferase assay in MDPC-23 cells treated with Cpne7-DP 1 µg, 10 µg or rCPNE7. (**d**) Effects of Cpne7-DP on the mineralized nodule formation of MDPC-23 cells in vitro analyzed by alizarin red S staining and corresponding semi-quantification. (**e**) Histological analysis of the subcutaneously transplanted 100 mg HA/TCP particles alone or with Cpne7-DP or rBMP2 in a 0.5% fibrin gel into immunocompromised mice for 12 weeks. Samples were stained with hematoxylin-eosin. Arrows indicate dentinal tubule-like structures; empty arrows indicate lacunae-containing osteocyte-like cells. All values represent the mean ± standard deviation of triplicate experiments. * *p* < 0.05 compared with control. GAPDH, glyceraldehyde 3-phosphate dehydrogenase.

**Figure 4 materials-13-04618-f004:**
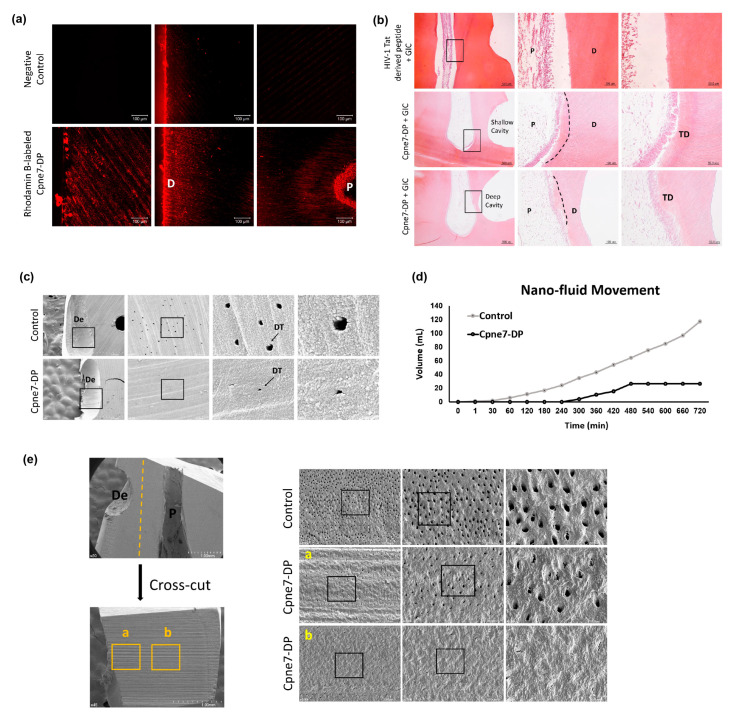
Cpne7-DP treatment induces the regeneration of tubular orthodentin in both dentin defect canine models, and promotes dentinal tubule occlusion by inducing peritubular dentin formation. (**a**) Evaluation of dentinal tubule penetration of Rhodamin B-labeled Cpne7-DP by using confocal laser scanning microscopy. (**b**) Histological analysis of dental pulp responses at the shallow cavity and deep cavity preparation areas treated with HIV-1 Tat-derived peptide (Control) or Cpne7-DP after 3 weeks by hematoxylin and eosin staining (*n* = 12). Dashed lines indicate the remaining original dentin/newly formed dentin interfaces. (**c**) Scanning electron microscopic analysis of defect surfaces after removal of the filling materials. Arrows indicate the dentinal tubules. (**d**) Nano-fluid movement (microleakage) analysis. (**e**) Scanning electron microscopic analysis of cross-sectioned surface between the defect and the pulp space. Areas noted by a and b indicate sectioned surfaces underneath the unaffected part of the crown and underneath the defect, respectively. P, pulp; D, dentin; TD, newly formed tertiary dentin; OD, osteodentin; De, defect; DT, dentinal tubules.

**Figure 5 materials-13-04618-f005:**
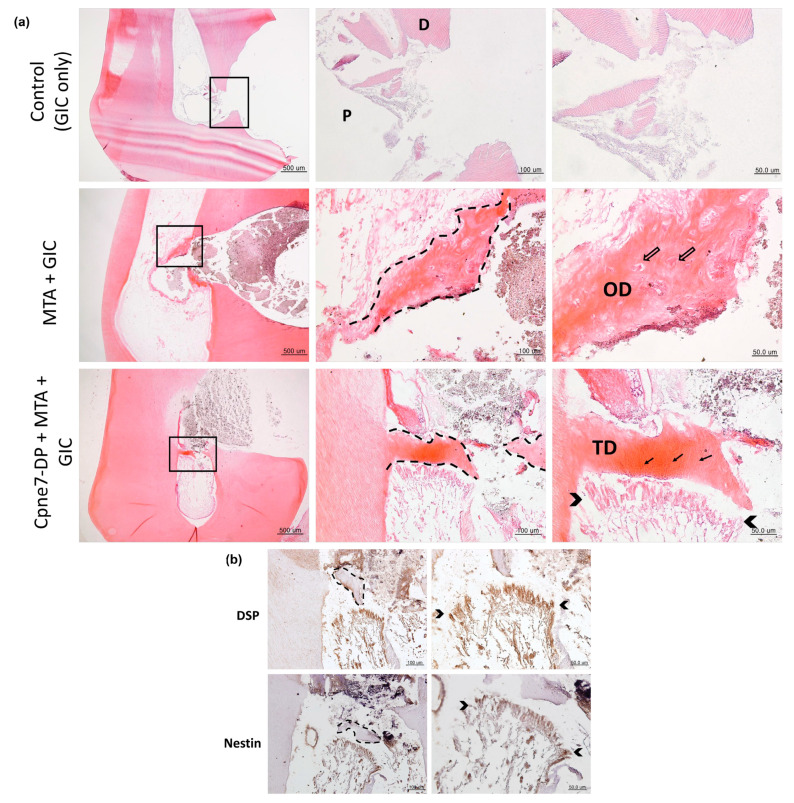
Cpne7-DP treatment induces the regeneration of tubular orthodentin in pulp exposure canine models. (**a**) Histological analysis of dental pulp responses at the complete pulp exposure areas filled with glass ionomer (GI) cement only, or sealed with MTA before GI cement filling, or sealed with MTA dissolved in Cpne7-DP before GI cement filling. Samples were analyzed by hematoxylin and eosin staining after 12 weeks (*n* = 6). (**b**) Immunohistochemical staining of cells underlying the newly formed dentin bridge in (A) with anti DSP (1:100) and anti-Nestin (1:100) in the pulp exposure model treated with Cpne7-DP. Empty arrows indicate the lacunae containing osteocytes; arrows indicate the dentinal tubules. Dashed line indicates the newly formed dentin bridge edge. Brackets outline the newly differentiated odontoblast layer. P, pulp; D, dentin; TD, newly formed tertiary dentin; OD, osteodentin.

**Figure 6 materials-13-04618-f006:**
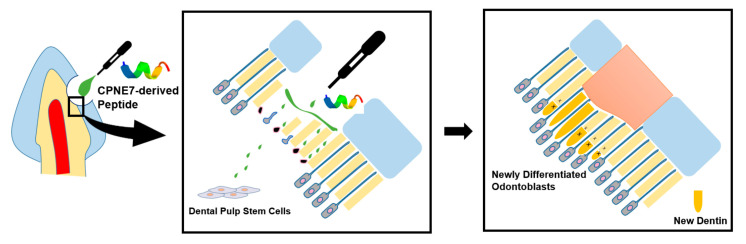
Schematic illustration of CPNE7 acting in two different ways to regenerate tubular dentin. In cases of a mild stimulus followed by shallow dentinal defects, treatment of CPNE7 leads to the reactivation of underlying resting-state mature odontoblasts to resume active matrix secretion. Under severe stimulus followed by deep dentinal defects that majorly damage underlying odontoblasts, treatment of CPNE7 induces differentiation of new odontoblasts from the pulp cell population, which can insert their cellular processes into the remaining dentinal tubules and form a tertiary dentin structure.

**Table 1 materials-13-04618-t001:** List of primers for real-time PCR.

Gene		Primer Sequence (5′–3′)
*Dentin sialophosphoprotein (DSPP)*	Forward	CAACCATAGAGAAAGCAAACGCG
	Reverse	TTTCTGTTGCCACTGCTGGGAC
*Dentin matrix protein-1 (DMP1)*	Forward	ACAGGCAAATGAAGACCC
	Reverse	TTCACTGGCTTGTATGG
*Nestin*	Forward	AGCCCTGACCACTCCAGTTTAG
	Reverse	CCCTCTATGGCTGTTTCTTTCTCT
*Glyceraldehyde 3-phosphate dehydrogenase (GAPDH)*	Forward	AGGGCTGCTTTTAACTCTGGT
	Reverse	CCCCACTTGATTTTGGAGGGA

**Table 2 materials-13-04618-t002:** List of antibodies for immunoassays.

Antigen	Source	Technique	Dilution	Molecular Weight (kDa)
Bone sialoprotein (BSP)	Produced as described previously (Lee et al., 2011)	IHC	1:200	35, 70
Dentin sialoprotein (DSP)	Produced as described previously (Lee et al., 2011)	WB, IHC	1:2000, 1:100–200	55
GAPDH	Santa Cruz	WB	1:5000	37
NESTIN	Millipore	WB, IHC	1:1000, 1:200	200
	Thermofisher	IHC	1:100	200

**Table 3 materials-13-04618-t003:** List of reagents and plasmids used.

**Reagents**	**Source**
Chlorpromazine	Sigma-Aldrich
Methyl-β-cyclodextrin	Sigma-Aldrich
Ethylisopropyl Amiloride	Sigma-Aldrich
Recombinant CPNE7	Origene
**Plasmids**	**Source**
Control siRNA	Ambion
Nucleolin-targeting siRNA	Ambion

**Table 4 materials-13-04618-t004:** Summary of the in vivo canine model results.

	Shallow Cavity	Deep Cavity	Pulp Exposure	
Cpne7-DP Treated	Tubular Dentin	Tubular Dentin	Incomplete Tubular Dentin	
Untreated (Control)	No New Hard Tissue	No New Hard Tissue	No New Hard Tissue	Osteodentin (MTA treated)

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
