# Peer review of "Tubular Dentin Regeneration Using a CPNE7-Derived Functional Peptide"

_materials, 2020, doi:10.3390/ma13204618_

Round 1

Reviewer 1 Report

This manuscript presents the tubular dentin regeneration by CPNE-7 derived functional peptide. This topic is highly relevant and the CPNE7-DP might be very promising material for conservative dentistry. However, there are some issues that need to be addressed.

  1. I would like to ask how deep were the deep preparations in the dogs molars?
  2. Where the animals anesthetized or sedated? How?
  3. It would be beneficial to explain the abbreviations used in the materials and methods section such as: rCPR7, NCDMP
  4. In the results section, you are writing about the possibility of restorative materials affecting the action of CPNE-DP. Composite resin and zinx oxide-eugenol are mentioned, wouldn't it be beneficial to add this information to the materials and methods sections?
  5. Why HIV-1Tat-derived cell-penetrating peptide was used as a control in peptide functional analysis and regeneration of tubular dentin analysis?

Author Response

1. I would like to ask how deep were the deep preparations in the dogs molars?

 Thank you for your comment. For deep cavity models, the drilling was stopped at the point where the underlying pulp started to show through as bluish-grey color.

2. Where the animals anesthetized or sedated? How?

 Thank you for your comment. The beagle dogs were anesthetized by licensed veterinarians at SNUSD with Zoletil 50(0.1-0.14ml/kg) and Rompun (0.01ml/kg) for total duration of 60 mins. 

3. It would be beneficial to explain the abbreviations used in the materials and methods section such as: rCPNE7, NCDMP

 Thank you for your comment. We followed your advice and added the information in the materials and methods section (2.1). 

4. In the results section, you are writing about the possibility of restorative materials affecting the action of CPNE-DP. Composite resin and zinx oxide-eugenol are mentioned, wouldn't it be beneficial to add this information to the materials and methods sections?

 Thank you for your comment. We followed your advice and added the information in the materials and methods section (2.1). 

5. Why HIV-1 Tat-derived cell-penetrating peptide was used as a control in peptide functional analysis and regeneration of tubular dentin analysis?

 Thank you for your comment. In order to rule out the possibility that any cell-penetrating peptide itself can stimulate odontoblasts and induce tubular dentin formation, we used one of the well-known cell-penetrating peptides as a control. 

Reviewer 2 Report

The study is interesting and was conducted with in-depth methodologies and analyzes; The use of CPNE7 and its derivative can aid in dentinal regeneration and pulp-dentinal protection following caries or trauma.

Only a few suggestions are to be made

  • Indicate the characteristics of the data extracted from the tests performed, and possibly represent and summarize them in tables
  • What is the mechanism by which CPNE7 or its peptide derivative, Cpne7-DP leads to tubular dentin regeneration and induces a reduction in sensitivity? , could the mechanism of action be represented through an image?
  • What are the limitations of this study?
  • What future applications can CPNE7 have. ? What can be the next research steps?

Author Response

1. Indicate the characteristics of the data extracted from the tests performed, and possibly represent and summarize them in tables

Thank you for your comment. We followed your advice and summarized the characteristics of the data in a table (Table 4).

2. What is the mechanism by which CPNE7 or its peptide derivative, Cpne7-DP leads to tubular dentin regeneration and induces a reduction in sensitivity? , could the mechanism of action be represented through an image?

 Thank you for your comment. We have attached a more detailed schematic illustration of the action mechanism of CPNE7 and Cpne7-DP, which we had decided to replace with a simpler version in the actual manuscript. As described in the figure 6, we believe that in case of mild stimulus followed by shallow dentinal defects, treatment of CPNE7 leads to the reactivation of underlying resting-state mature odontoblasts to resume active matrix secretion. Under severe stimulus followed by deep dentinal defects that majorly damage underlying odontoblasts, treatment of CPNE7 induces differentiation of new odontoblasts from pulp cell population, which can insert their cellular processes into the remaining dentinal tubules and form tertiary dentin structure.

3. What are the limitations of this study?

 Thank you for your comment. We added some limitations (including exact action mechanism of Cpne7-DP unknown, lack of quantification of tertiary dentin formed, lack of information about additive effects of topical application) in the Discussion part. 

4. What future applications can CPNE7 have? What can be the next research steps?

 Thank you for your comment. Few of the further studies we are focusing on right now include the effects of CPNE7 in stopping the demineralization process in Rat dental caries model, its function in autophagic activity and morphological differentiation of odontoblasts. 

Reviewer 3 Report

Dear authors,

I am glad to be chosen as a reviewer to your interesting paper. Great work has been done. Congratulations. Still, there are few concerns about your paper as follows:

Abstract: You wrote this part in a too general way. Try to be more specific.

"CPNE7 and Cpne7-DP" What is the difference?

"for oral disease-targeted application" What kind of disease are you referring? be more specific.

Introduction: Very nice and clear written. 

I suggest you to introduce a paragraph about the application of

Cpne7-DP in oral disease as you have referred in the abstract. Do you think this one can be applied only on healthy subject? What about in those with oncological treatment?

M&M:

Why did you choose beagle dogs as an experimental animal?

Did you calculate the sample size?

Why you did not use an IHC analysis?

Results:

Well written and presented. 

You should insert the figures into the main document as stated by the rules of the journal.

Discussion:

"In addition to causing hypersenstivity, exposed dentinal tubules can be a route for the invasion of foreign substances" - What foreign substances?

"Although the efficiency of hard tissue regeneration differs in model animals, regeneration studies in large mammals (e.g., dogs) have been reported to be more challenging than small animals (e.g., mouse)" - What about human studies? Do we have a comparison?

Write about the strengths and downs of your research.

Conclusion: This chapter is too long. Try to be more specific and shorter.

 Figures and tables: very well written and structured. No further improvements are needed here.

Author Response

1. Abstract: You wrote this part in a too general way. Try to be more specific.

 Thank you for your comment. We followed your advice to be more specific, and changed the Abstract structure to the one that has sub-sections of Objective, Methods, Results, Conclusion, and Clinical Significance. 

2. "CPNE7 and Cpne7-DP" What is the difference?

 Cpne7-DP is a 10-mer functional peptide derived from protein Copine7 (CPNE7). 

3. "for oral disease-targeted application" What kind of disease are you referring? be more specific.

 Thank you for your comment. We followed your advice and changed it to "treatment of dentinal loss and sensitivity."

4. Introduction: I suggest you to introduce a paragraph about the application of Cpne7-DP in oral disease as you have referred in the abstract. Do you think this one can be applied only on healthy subject? What about in those with oncological treatment?

 Thank you for your comment. We followed your advice and added the relevant information about dental caries in the first paragraph of the Introduction. As of now, we have no evidence of possible therapeutic effects of Cpne7-DP in oncological treatment, However, we appreciate your question and will be willing to carry on further research. 

5. M&M: Why did you choose beagle dogs as an experimental animal?

Did you calculate the sample size?

Why you did not use an IHC analysis?

 As mentioned in the 5th paragraph in the Discussion, large mammals provide a more appropriate model for mimicking human disease, and are thus more transferable to a human model. That is why we chose beagle dogs instead of mouse or rats for experimental animal. 

 For in vivo canine studies, a total of 10 beagle dogs (2 y) were operated on, and while exact numbers for each experiment are included in the figure legends, from at least 6 premolars up to 12 premolars were assigned to every control and experimental group.

 While Figure 5 (pulp exposure model) has an IHC result for DSP and Nestin, our previous study (Park et al., CPNE7 Induces Biological Dentin Sealing in a Dentin Hypersensitivity Model (2019), Journal of Dental Research) presented IHC results for indirect pulp capping model. 

6. Results: You should insert the figures into the main document as stated by the rules of the journal.

 Thank you for your comment. We followed your advice and moved all the main figures to the relevant parts in the main document. 

7. Discussion: "In addition to causing hypersenstivity, exposed dentinal tubules can be a route for the invasion of foreign substances" - What foreign substances?

 Thank you for your comment. We added the sentence "Exposed dentinal tubules in dental caries serve as a route for bacterial invasion that may result in various pulpal responses." in the Introduction part for better understanding.

"Although the efficiency of hard tissue regeneration differs in model animals, regeneration studies in large mammals (e.g., dogs) have been reported to be more challenging than small animals (e.g., mouse)" - What about human studies? Do we have a comparison?

 Thank you for your comment. Unfortunately, we could not find a comparison with human studies in the article we referred to. 

Write about the strengths and downs of your research.

 Since we believe strengths of our research are mentioned several times throughout the manuscript, we added some limitations (including exact action mechanism of Cpne7-DP unknown, lack of quantification of tertiary dentin formed, lack of information about additive effects of topical application) in the Discussion part. 

8. Conclusion: This chapter is too long. Try to be more specific and shorter.

 Thank you for comment. We agreed that the Conclusion was pretty much the repetition of the Abstract, and deleted it. Instead, we made a sub-section Conclusion as a part of the Abstract. 

Reviewer 4 Report

1) in the abstract and conclusions, the beginning of the phrase "Our results indicate that ..." should be replaced with a brief listing of the results obtained. This should be done in order to reduce the time spent by readers searching for new scientific information.

2) in the fourth section "Discussion "is not correct use of the concept of "optimal". There was no mathematical model and the optimization method was not used, so we need to talk about the rational or best of the options.

Author Response

1. in the abstract and conclusions, the beginning of the phrase "Our results indicate that ..." should be replaced with a brief listing of the results obtained. This should be done in order to reduce the time spent by readers searching for new scientific information.

 Thank you for your comment. We followed your advice and added more detailed results in the abstract, while deleting the repetitive Conclusion part.

2. in the fourth section "Discussion "is not correct use of the concept of "optimal". There was no mathematical model and the optimization method was not used, so we need to talk about the rational or best of the options.

Thank you for your comment. We followed your advice and deleted the "optimal uses" in the Discussion.  

Round 2

Reviewer 2 Report

The authors followed all the changes requested and answered all the doubts raised about the manuscript. I consider the manuscript worthy of publication